# UV-B Physiological Changes Under Conditions of Distress and Eustress in Sweet Basil

**DOI:** 10.3390/plants8100396

**Published:** 2019-10-04

**Authors:** Haana Mosadegh, Alice Trivellini, Mariella Lucchesini, Antonio Ferrante, Rita Maggini, Paolo Vernieri, Anna Mensuali Sodi

**Affiliations:** 1Institute of Life Sciences, Scuola Superiore Sant’Anna, Pz. Martiri della Libertà 33, 56127 Pisa, Italy; hana_mosadegh@yahoo.com (H.M.); a.mensuali@santannapisa.it (A.M.S.); 2Department of Agriculture, Food and Environment, Via del Borghetto 80, 56124 Pisa, Italy; mariella.lucchesini@unipi.it (M.L.); rita.maggini@unipi.it (R.M.); paolo.vernieri@unipi.it (P.V.); 3Department of Agricultural and Environmental Sciences, Università degli Studi di Milano, I-20133 Milano, Italy; antonio.ferrante@unimi.it

**Keywords:** irradiation, photodamage, photosynthetic pigments, plant hormones

## Abstract

UV-B radiation has been previously reported to induce protective or deleterious effects on plants depending on the UV-B irradiation doses. To elucidate how these contrasting events are physiologically coordinated, we exposed sweet basil plants to two UV-B doses: low (8.5 kJ m^−2^ day^−1^, 30 min exposure) and high (68 kJ m^−2^ day^−1^, 4 h exposure), with the plants given both doses once continuously in a single day. Physiological tests during and after both UV-B exposures were performed by comparing the stress-induced damage and adverse effects on photosynthetic activity, the concentration and composition of photosynthetic and non-photosynthetic pigments, and stress-related hormones biosynthesis in basil plants. Our results showed that upon receiving a high UV-B dose, a severe inactivation of oxygen evolving complex (OEC) activity at the PSII donor side and irreversible PSII photodamage caused primarily by limitation of the acceptor side occurred, which overloaded protective mechanisms and finally led to the death of the plants. In contrast, low UV-B levels did not induce any signs of UV-B stress injuries. The OEC partial limitation and the inactivation of the electron transport chain allowed the activation of photoprotective mechanisms, avoiding irreversible damage to PSII. Overall results indicate the importance of a specific response mechanisms regulating photoprotection vs irreversible photoinhibition in basil that were modulated depending on the UV-B doses.

## 1. Introduction

The formation of a stratospheric ozone layer has established a kind of solar radiation filter to absorb completely harmful wavelengths such as UV-C (<280 nm) and much of the UV-B (280–315 nm) emitted by the sun. During the 20th century an increase in UV-B radiation was detected to reach the earth’s surface, caused by the depletion of the ozone layer triggered by chlorofluorocarbons release [1]. Currently, however, predictions suggest the beginnings of a recovery of stratospheric ozone, with some areas that will likely show a decrease of UV radiation to levels lower than those before the 1960s, while other regions will experience enhanced levels of UV-radiation [2]. In the light of these new indications, UV-B radiation effects on plant behaviors are of increasing interest in the plant science community. In fact, considering that solar radiation is the primary energy source for plants, understanding the mechanisms which might regulate UV-induced injurious and protective responses could potentially lead to improved plant productivity and quality. Due to its high energy, UV-B is potentially deleterious to many organisms, including plants. An enhancement of UV-B can determine a diverse range of plant responses depending on the fluence rate, dose, duration, and wavelength of the UV-B treatment, though generally in the literature, plant UV-B responses are defined by low or high dose radiations [3,4,5,6,7,8,9,10]. The ecologically relevant low UV-B radiation around the ambient level can act as an important environmental cue by inducing regulatory effects on plant morphology, physiology, and biochemistry to stimulate UV-B protection mechanisms and/or amelioration of UV-B damage, and these effects do not impede necessarily growth and development of plants [2,4,11,12]. On the other hand, a high dose of UV-B, above ambient level, may cause exceptionally high biological damage to the photosynthetic organism, leading to severe UV-B distress with a massive production of ROS, damage to DNA, proteins, and membranes, the inhibition of protein synthesis, destruction of photosynthetic pigments and photosynthetic reactions which can induce programmed cell death (PCD), and lastly the death of plants [13,14,15,16].

The main target of UV-B radiation in photosynthetic organisms, since exposure to such high energy light causes inhibition of the photosynthetic machinery, is represented by the PSII complex [17]. However, the vulnerable targets and the physiological responses in UV-induced damage in PSII (PSII photodamage) have been difficult and controversial to explain. Previous reports suggested that the primary cause for the UV-induced inhibition of the PSII function is the blockage of electron transport water-oxidizing complex (OEC) at Mn cluster on the donor side, while others reported evidence for inactivation at quinone electron acceptors (QA and Q B) on the acceptor side [18,19].

Hence, the present study was conducted to probe and compare the protective versus deleterious effects of two different UV-B doses, low (8.5 kJ m^−2^ d^−1^) and high (68 kJ m^−2^ d^−1^), considering the differential responses of the photosynthetic machinery of sweet basil (*Ocimum basilicum* L.). With this aim, chlorophyll fluorescent transients followed by the JIP-test, concentration and composition of photosynthetic and non-photosynthetic pigments, and stress-related hormones biosynthesis were evaluated. Sweet basil is one of the most economically important cultivated herb belonging to the *Lamiacae* family, which has been extensively used by many cultures worldwide as a culinary herb and also has been exploited by cosmetic, food, and pesticides industries for its high natural content in polyphenols [20]. A deeper understanding of the temporal patterns involved in different UV-B responses will help to dissect protective from irreversible damage-related events, and may potentially lead to improved agronomic crop performance and industrial applications.

## 2. Results

### 2.1. Visual Effects of UV-B Exposures

Sweet basil plants grown in a “*in vitro-vivo* system”, which was recently described [10], were UV-B irradiated using two different UV-B doses, low and high dose, in order to depict the overall effects of UV radiations on photosynthesis using parameters derived from chlorophyll *a* fluorescence measurements, on physiological (hormones) and biochemical (photosynthetic pigments and phenolic acids) traits. In the low UV-B dose model, sweet basil plants were UV-B irradiated for 30 min, obtaining a daily UV-B dose flux of 8.5 (kJ m^−2^ d^−1^). The UV-B dosage experienced by these sweet basil plants is slightly higher compared to the UV-B regimes (2.3–7 kJ m^−2^ d^−1^) encountered at mid latitudes (35°–45°N) during the growing period (spring season) in the northern hemisphere [21,22]. In the other model, the plants were UV-B irradiated for 4 h reaching a fluence rate of 68 kJ m^−2^ d^−1^, which is a high UV-B dose. After the UV-B exposures, plants subjected to the two different experimental models were placed in a light regime along with the already present control plants for up to 72 h (recovery period). An overview of experimental design regarding the conditions of plant growth, UV-B irradiation, and recovery is reported in Figure 1. Glossy-light green leaves were observed after 24 days of high UV-B treatment, and leaf-browning and necrosis/wilting were also induced later (over 48 and 72 h) by a high UV-B dose (Figure 2A). Glossy leaves were not observed after low UV-B irradiation, and the treated plants appeared to be similar to the control ones until 48 h was reached. Then, the basil plants exhibited leaf curling at the end of recovery period (Figure 2). At the end of recovery period (72 h) a staining with Evans Blue, a dye that is readily taken up specifically by dead cells was used to confirm the involvement of high UV-B dose induced cell death in *O*. *basilicum* leaves. Positive staining was clearly observed in cells of leaves exposed to high UV-B radiation [Figure 2B]. In contrast, no Evans Blue staining was observed in cells of leaves exposed to low UV-B radiation, similarly to in leaves not exposed to UV-B (control).

### 2.2. The Impact of UV-B Radiation on PSII Photochemistry

The effect of high and low UV-B dose on the photosynthetic machinery of *O*. *basilicum* leaves was investigated by measuring the fast Chlorophyll a fluorescence transient at 0, 24, 48, and 72 h after UV-B exposures (low and high doses). The mean values of measured and calculated fluorescence parameters [Table 1, Table 2], suggest possible differences in energy fluxes at the donor as well as at the acceptor side of PSII [23,24].

Compared with controls, leaves exposed to high UV-B radiation showed a significant decrease in the maximal fluorescence intensity (Fm) and in the efficiency of the water-splitting complex on the donor side of PSII (Table 2) at almost any time after UV-B exposure, whereas the initial fluorescence (Fo) was much higher (Table 2). The increase of Fo was observed at a high UV-B dose already after 24 h of exposure. Moreover, the effects of high UV-B radiations on the inactivation of the electron transport at the donor side of PSII were noticed through a significant increase over recovery time in the Fk/Fj ratio, Vj, and Vk (Table 2). Immediately after high UV-B radiations (0 h), time needed to reach Fm (Tfm) and energy necessary for the closure of all reaction centers (Sm) highly increased and then gradually decreased before drastically dropping after 72 h of UV-B exposure. Furthermore, leaves exposed to high UV-B dose demonstrated a more rapid accumulation of closed RCs, with increasing Mo values at 48 and 72 h.

UV-induced changes were studied on the basis of flux ratios like ϕP0, ϕE0, and ΨE0, which affect electron transport at the acceptor side of PSII, and δR0, δD0, and ϕR0, which are indirectly associated with PSI. The quantum yield of primary photochemistry (ϕpo) decreased significantly, when plants were exposed to high UV-B doses from 24 h to 72 h, and both the quantum yield for electron transport (ϕEo) and the efficiency per trapped excitation (Ψo) showed almost comparable changes (Table 2). The efficiency with which an electron can move from the reduced intersystem electron acceptors to the PSI end-electron acceptors (δR0), promptly increased after high UV-B exposure (at 0 h) but at the end of the recovery time (72 h) the value was strongly reduced compared to the control and low UV-B exposure. Compared to the control, high UV-B exposed leaves had a significantly decreased quantum yield for the reduction of end acceptors of PSI per photon absorbed (ϕR0) after 48 and 72 h.

### 2.3. Overall Effects of UV Radiations on Photosynthetic Pigments

The chlorophyll and related pigments such as carotenoids were measured since these molecules have important role in leaf functionality, and are involved in light perception and protection against photooxidation. To measure the changes in pigments concentration of leaves, HPLC analysis of isolated photosynthetic pigments during low and high UV-B radiations, as well as after these exposures over a period of 72 h, was carried out. The results are shown in Figure 3A–F. The overall changes in the levels of chlorophyll a and chlorophyll b showed a similar trend (Figure 3A,B). In leaves exposed to high UV-B radiation, the chlorophyll levels began to decrease after 24 h of recovery and still remained significantly lower compared to both control and low UV-B treated leaves over the 72 h recovery period (Figure 3A). Meanwhile, in control and low UV-B exposed leaves, the chlorophyll content was unaffected, resulting in a non-statistically significant difference of p < 0.005 between ctrl and low for both chlorophyll a and chlorophyll b.

A slightly different trend was observed for all carotenoids measured (Figure 3C–F). Again, in control leaves of sweet basil plants, the contents of violaxanthin, zeaxanthin, lutein and β-carotene did not change throughout the experimental time. On the other hand, the exposure of basil leaves to high UV-B dose led to a general decrease in all carotenoids tested. In detail, the violaxanthin and β-carotene contents were reduced 24 h after the treatment during the recovery period (Figure 3C,E). The content of zeaxanthin showed a decline during UV-B exposure starting from 30 min (Figure 3D), whereas lutein content decreased immediately after 4 h UV-B exposure (Figure 3F). Low UV-B dose exposure did not influence violaxanthin content (Figure 3C), showing a similar trend in untreated control plants (Figure 3C). A rapid and significant decrease of zeaxanthin (Figure 3D) was observed at the end of UV-B treatment (30 min) and after 4 h from the beginning of experiments. Subsequently, a transient increase at 24 h (slightly higher than in the control plants) and at 48 h was reported, and then at 72 h, the content dropped significantly compared to the value of control plants but showed a significantly higher content related to high UV-B treated plants. After 4 h from the beginning of the experiment, a generalized significant decrease of β-carotene in low UV-B exposed plants was observed and this reduction was prolonged during the entire recovery period, with no statistical differences compared to plants treated with high UV-B dose (Figure 3E). In low UV-B treated plants, the lutein content did not show any significant differences among the different time points compared to the control (Figure 3F).

### 2.4. Overall Effects of UV Radiations on Phenolic Acid Accumulation

Among the nine phenolic acids evaluated (of interest), only rosmarinic acid (RA) was found in considerable concentrations in all analyzed samples, while the other metabolites were present in trace quantities or below the detection limits (0.05 mg g^−1^ DW). The typical chromatogram of an extract of basil sample is reported in Figure 4A and showed one main peak that was identified as RA by means of LC-MS analysis. The rosmarinic acid (RA) content after 48 h drastically increased in UV-B treated plants exposed to high dose and its content remained significantly high until the end of experiment (Figure 4B). During the experiment, the RA content in control plants and in low-doses UV-B treated plants did not change among the different time points (Figure 3B).

### 2.5. Endogenous Hormones Were Affected Differently by Low and High UV-B Light Conditions

Endogenous content of ABA increased significantly in UV-B treated plants compared to the untreated ones after 15 min and 30 min during both (low and high) UV-B exposures. Then the high UV-B exposure led to an increase of ABA content at 24h and its level remained significantly higher compared to the control until the end of the experiment (Figure 5A). On the other hand, plant exposed to low UV-B did not change endogenous ABA content during the whole recovery period, in a similar manner of control plants. A biphasic response to UV-B has been observed for ethylene evolution (Figure 5B). In UV-B treated plants, there was a first high peak after 30 min, in both high and low UV-B exposure treatments (Figure 5B). Then, the evolution of ethylene again increased with a second peak after 6 h under low UV-B exposure with levels similar to the leaves exposed to high dose. However, in high UV-B dose exposed plants ethylene continued to increase reaching the highest peak after 24 h and then declined but maintained higher values compared to the control until 72 h (Figure 5B). In contrast, in low UV-B exposed leaves, the ethylene production did not show any significant differences from control ones from 24 to 72 h of recovery time (Figure 5B).

## 3. Discussion

Light is the essential component to drive photosynthesis, but it is intrinsically harmful to the photosynthetic machinery [27]. Exposure of crops to strong visible or UV light, results in light-induced inactivation of photosystem II (PSII), which may cause the death of the photosynthetic organism, when the rate of damage exceeds the rate of repair process [17]. Data presented in our study offer a clear picture of the two response modes, adaptation versus death, using two contrasting UV-B doses (low UV-B: 8.5 kJ m^−2^ day^−1^, and high UV-B: 68 kJ m^−2^ day^−1^), by comparing the stress-induced damage and adverse effects on photosynthetic activity (mainly related to PSII), concentration, and composition of photosynthetic and non-photosynthetic pigments, and stress-related hormones biosynthesis in sweet basil plants.

In our system, stress effects occurred mainly at high doses of UV-B which lead to the death of the plants at the end of the recovery period. The leaves of these plants were permeable to Evans blue staining and therefore were considered to be dead as their plasma membrane was no longer semipermeable. On the other hand, low levels of UV-B did not induce any signs of stress injuries. Under these conditions, high UV-B intensity was proposed to be associated with a state of distress leading to a ROS burst which resulted in PCD, whereas low UV-B intensity was considered as a sort of positive stress (eustress), allowing the plants to cope against adversity [9].

Under UV-B light, the photoinhibitory process is dependent on the time and fluence rate which can lead to partial or complete disintegration of chlorophyll protein complexes, resulting in an increase in ROS production and finally to death of the photosynthetic organism. Previous reports reviewed by Kataria et al. [18], showed that photoinactivation by UV light possibly affects two targets, the water-oxidizing manganese (Mn) cluster which results in the inactivation of the electron transport chain and alterations at the acceptor side of PSII. However, the results reported did not give a clear-cut indication as to which one was the primary cause for the UV-induced inhibition of the PSII function. In the present report, UV-B induced inhibition of the PSII function affected two targets in two distinct phases, first damage occurred in the water-oxidizing manganese (Mn) cluster and resulted in the inactivation of the electron transport chain, and then the limitation at the acceptor side of PSII was observed. This last event exerted a strong influence on the repair mechanism leading to plant death, due to harmfully high UV-B dose, according to the unified model of photodamage recently proposed by Zafer et al. [19]. These authors showed that the loss of activity of the Mn cluster is attributed to the first event of photodamage, and it is a consequence of photodamage, since the addition of electron acceptors slowed down the loss of PSII activity, clearly indicating the acceptor side limitations as an effective cause of photoinactivation under high light illumination. In the first phase (immediately after UV-B exposure), only high UV-B exposed leaves manifested severe PSII damage through the primary inactivation oxygen-evolving complex (OEC) activity inhibiting the electron transfer at the donor side. Damage, limitation, and inactivation of the OEC was assessed through the increase in relative variable fluorescence at 300 µs (K-step), Vk [28] and Fk/Fj ratio as a quantitative measure for the inactivation of the PSII donor side [29]. The Fv/Fo ratio was also used to evaluate the efficiency of the water-splitting complex on the donor side of PSII [30], where a decrease of the value of Fv/Fo has been clearly associated with damage caused by light stress [31]. Under a high UV-B dose, the OEC was the most UV-B sensitive component of PSII. The Fk/Fj ratio increased immediately after high UV-B exposure and continued to increase markedly during recovery, indicating that UV-B stress inactivated the OEC activity. Similarly, the prompt and progressive decrease in Fv/Fo indicated that the water-splitting system of the donor side of PSII might be seriously impacted by high UV-B radiation, leading to OEC inactivation. From the data calculated by JIP-test and shown in Table 2, a slight and retarded alteration in Fk/Fj and Fo/Fv ratios and no significant change in Vk after low UV-B dose, indicated that OEC was less affected than with high UV-B dose and that the electron transport from PSII donor side to PSII reaction center was partly limited. Under this condition (i.e., basil plants exposed to low UV-B dose), it appears clear that low UV-B dose did not lead to PCD, conversely to what was observed under high UV-B doses (Figure 2), which resulted in the death of plants at the end of the recovery period. This was possibly caused by reactive oxygen species (ROS) generation and downstream damages to biomolecules, such as oxidation of lipid and protein, and DNA, as well as an enhancement in lipid peroxidation [32,33,34].

However, the cause of irreversible photoinactivation observed under high UV-B was attributed to a second and later phase during recovery (from 24 to 72 h), which involved the limitations of the acceptor side and greatly retarded electron transfer between QA- and QB, showing a strong influence on the loss of PSII efficiency (Fv/Fm), as previously reported by Mosadegh et al. [10]. In fact, when UV-B-induced changes (i.e., adaptation in basil plants exposed to low UV-B dose and death in basil plants exposed to high UV-B doses) were compared on the basis of the flux ratio (Table 2), only the highest UV-B dose significantly affected electron transport parameters at the acceptor side of PSII. ϕP0 corresponds to the maximum quantum yield of primary photochemistry. A decrease in ϕP0 under high UV-B dose indicated that the trapping probability of the reaction center (RC) was blocked, decreasing the rates of electron transport beyond QA-, and thus the photosynthetic efficiency was inhibited. Similar reductions were observed in the quantum yield of electron transport ϕE0. ΨE0 represents the probability of a trapped exciton moving an electron further than QA-, thus it is the efficiency of electron transfer from QA- to QB [23]. The high UV-B radiation caused a decrease in ΨE0, ϕE0, and ϕP0, reflecting a less efficient electron transfer after QA- to QB. Moreover, the increase in both Mo (which represents the net rate of closed RCs accumulation) and Vj (which reflects a measure of the fraction of the primary quinone electron acceptor of PSII in reduced state) further confirmed that high UV-B exposure could result in QA- accumulation. The damage to the PSII inflicted by UV-B radiations appears to be initially located at the water-oxidizing Mn cluster. However, irreversible photodamage of PSII leading to cell death was primarily caused by limitations of the acceptor side under high UV-B radiation.

Irradiation with high UV-B destroyed photosynthetic pigments through inhibition of their biosynthetic pathways and downregulation of genes associated with photosynthesis is causing its subsequent inhibition [35,36,37]. Chlorophyll degradation is the most common form of damage that can be observed after UV exposure [38,39] with consequence on the crop recovery. Similarly, high UV-B dose exposure in basil induced inhibition of photosynthesis resulting from chlorophylls degradation (Figure 3A,B). In addition, carotenoids contents were also significantly reduced in basil plants subjected to higher UV-B irradiation, and their reduction was previously reported to have serious consequences on chlorophyll protection from photoxidative destruction [18]. Lutein and other xanthophylls, constituents of the photosynthetic protein pigment LHCII complexes, are essential in photoprotection mechanisms to avoid the light-induced damage of the photosynthetic apparatus due to the formation of ROS under excess light [40]. A lower level of lutein, zeaxanthin and a significant decrease in violaxanthin availability were observed in high UV-B dose treated plants and may result from a serious degradation of LHCII. This effect was probably caused by the excessive and inefficient excitation energy transfer from antenna to the PSII RC through the physical separation of the PSII RC from associated pigment antennae, and the inhibition of electron flow reflecting the accumulation of the reduced form of QA- [41,42]. According to this effect, the increase in Fo was observed only in basil plants exposed to high UV-B. The increase of Fo in high UV-B model indicated that PSII was already impaired after 24 h as well as the electron transfer chain. In contrast, basil plants exposed to low UV-B dose (8.5 kJ m^−2^ day^−1^) did not show any effect on the photosynthetic pigments such as chlorophylls and lutein; however, the levels of zeaxanthin and beta-carotene were slightly reduced by UV-B suggesting the involvement of protective or repair mechanisms that can be modulated by this light quality. Consistently, the decrease in Fm may indicate a sustained engagement of zeaxanthin for energy dissipation, and thus stimulation of a photoprotective mechanism known as the xanthophyll cycle [43]. In addition, the decreased retention of β-carotene after low UV-B treatment might be useful during repair of PSII, since β-carotene seems to be released from the reaction center and has been shown to be essential for the assembly of PSII [44,45,46].

Another mechanism of photoprotection from UV-B radiation is the activation of the secondary metabolism and in particular, the phenylpropanoids pathway, which induces the biosynthesis of a wide range of phenolic compounds [47]. The increase of total phenolic content was previously observed in basil showing a UV-B dose-response trend [10]. These molecules with their conjugated double bonds can remove the ROS concentration reducing the cell damage. The accumulation of these molecules usually occurs during and after the UV exposure [48]. In our experiments, rosmarinic acid (RA), the main phenolic compound contained in the tissues of several plant species belonging to the *Lamiaceae* [20] was present at high concentrations only in basil plants exposed to high UV-B dose during 48 and 72 h of recovery. In agreement with a previous work on different basil growing systems [49], apart from RA, no other caffeic acid derivates were detected at a significant level in any of the analyzed samples. In basil, different UV-B doses might differentially modulate pathways between non-photosynthetic and photosynthetic pigments, playing an important antioxidant role in the photoprotection. A profound effect of oxidative stress which may be related to severe ROS overproduction has been shown to strongly elicit the biosynthesis of RA as reviewed in Trivellini et al. [20] and reported in Dewanjee et al. [50], and this situation probably occurred in basil exposed to high UV-B dose. In contrast under a low UV-B dose, light-induced damage of the photosynthetic apparatus could be ameliorated with the sustained engagement of the xanthophyll cycle as discussed above.

Effects of UV-B on plants are largely dependent on the control of hormonal pathways and as reviewed in Vanhaelewyn et al. [51] can be roughly classified as photomorphogenic effects and stress effects. In the present study, the modulation of hormonal pathway affecting the biosynthesis of ABA and ethylene was monitored during and after high and low UV-B exposure to provide a more detailed and comprehensive picture of the role of these hormones in injured/died and acclimated basil plants. ABA and ethylene are widely known as stress phytohormones, and their concentrations change facilitating plants to survive and grow well under a broad range of stress conditions [52,53,54,55,56,57]. The mechanisms controlling hormone dose-dependent protective/injurious response are largely unexplored in UV-B stress. When basil plants were exposed to low UV-B radiation, a rise in ABA accumulation occurred as a first and active response after 15 min, ethylene evolution showed a biphasic time course with its enhancement after 30 min and occurred concomitantly with the increased endogenous level of ABA, which continued after 6 h in the recovery process. On the other hand, in response to high UV-B dose, both ABA and ethylene considerably changed their concentrations, reaching the highest values during recovery. These behaviors might suggest differential phytohormone-regulative responses to UV stress which depend on hormonal concentrations: protective (at relatively low concentration) vs injurious (at high concentration) responses. An endogenous increase of ABA has been reported to cause a photoprotective effect as ABA biosynthesis mutants are more sensitive to UV-B [58]. Moreover, exogenous application ameliorates the impact of excessive excitation energy on PSII, through the de-epoxidation of xanthophyll-cycle components and preserves the photosynthetic pigments [59,60]. The increase of ABA after 6 h up to 72 h can be also induced by the degradation of carotenoids in particular violaxanthin, which contributes to the indirect ABA biosynthesis route. Regarding ethylene, few data have been published on the interaction between this hormone and UV-B stress. UV-B radiation mainly at high intensities stimulates ethylene biosynthesis acting as promoter of cell death in ROS-dependent PCD [59,60,61,62,63]. However, a physiologically relevant concentration of exogenously applied ethylene has been reported to regulate protection of photosynthesis against Ni- and Zn-induced heavy metal stress in *Brassica juncea* plants [64]. Therefore, taking into account that both ABA and ethylene are a part of a UV-B stress response mechanism that is also involved in the response of photosynthetic machinery to abiotic stresses [56], this study proposes that their different level of concentration in response to low and high UV-B dose could potentially be a sensitive regulator of appropriate physiological switching between protection mechanisms and PCD propagation.

In conclusion, our studies showed how basil plants respond differently to high and low UV-B irradiations by exploring the behavior of PSII activity, photosynthetic and non-photosynthetic pigments, and stress-related hormones as a targets, thereby deducing precise physiological mechanisms controlling protective vs injurious/destructive/harmful responses. There are different mechanisms of UV-B responses acting in basil as a function of the time and fluence rate (low and high). Low UV-B dose induces a sort of positive stress (eustress) through a partial limitation of OEC and the inactivation of the electron transport chain. This condition allows the activation of photoprotective mechanisms to counteract ROS and avoid irreversible damage to PSII. High UV-B dose leads to a state of catastrophe which starts with the primary and severe inactivation of OEC activity inhibiting the electron transfer at the PSII donor side, and finally shifts to irreversible photodamage of PSII caused by a limitation of the acceptor side. High UV-B intensity overloads the protective and repair mechanisms of photosynthetic protein pigment complexes and causes oxidative stress manifested with a peak in RA and subsequent ROS burst resulting in PCD. Basil plants exposed to low and high UV-B intensity differ in their ABA and ethylene accumulation, which could act as a sensitive regulator of appropriate physiological switching between protection mechanisms and PCD propagation, showing a relatively low concentration during low UV-B dose exposure for both hormones, and a high hormonal increase during recovery after high UV-B dose exposure.

## 4. Materials and Methods

### 4.1. Plant Material and Growing Condition

Seeds of *Ocimum basilicum* L. (cv. Genovese) were obtained from GBIS/I (Genebank Information System of the IPK Gatersleben, Germany). Seeds were surface sterilized by soaking in a solution of sodium hypochlorite 20% plus 0.01% Triton X-I00 for 10 min and rinsing four times with sterile water. The seeds were plated on MS agar media (Murashige and Skoog, 1962) with 3% (w/v) sucrose and 1.2% (w/v), Gelrite™ (Duchefa, NL), pH 5.7. Seeds were put at 4 °C for 48 h to synchronize germination. Then the plates were transferred under a 16 h photoperiod, at a constant 22 °C with a light level of 100 μmol m^−2^ s^−1^. Seven days old seedlings were transferred on Falcon™ (BD Bioscience, Belgium) tubes including agarized MS medium without sucrose and growth regulators. The Falcon containers were suitably modified to allow the growth of two individual basil plantlets with root apparatus inside the “in vitro” medium and the aerial part “in air” condition to avoid interference of the plastic lids with the UV-B treatments [Figure 1] [10]. Basil-plants grown in tubes for two weeks were used for all experiments. Plants were grown under a 16 h photoperiod, at a constant temperature of 22 °C and 65% humidity. Cool white light in the growth chamber was provided by L 36 W/76 florescence tubes (Osram, Munich, Germany). The light level was measured using a Field Scout ™ Light Sensor Reader equipped with a Quantum Light Sensor (Spectrum Technologies, Inc. Painfield, Illinois USA) and was 100 μmol m^−2^ s^−1^ during seedling growth and after irradiation (recovery stage). A spectrum of the light emission provided by the Osram company is provided as Appendix A. The plants within the growth chamber were arranged randomly on one shelf.

### 4.2. UV-B Treatments

The UV-B-exposure was performed in absence of light, in the dark. For the UV-B light treatments was used a completely shielded cabinet placed inside the growth chamber. The cabinet walls were covered with a 6 mm black polyethylene sheeting to ensure a maximum light blockage from the growth chamber. In addition, the cabinet was equipped with UV-B lamps (UV-B GL20 SE, Sankyo Denki, Tokyo, Japan), whose emission spectrum ranged λ from 280 to 400 nm, resulting in a maximum peak at 325 nm. A spectrum of the UV-B light emission by the Sankyo Denki company is provided as Appendix A. The “fluence” (UV dose) expressed as kJ m^−2^ d^−1^ was obtained by the “fluence rate” (µmol m^−2^ sec^−1^) measured with a Light Sensor Reader equipped with a UV Sensor (Field Scout ™ Spectrum Technologies, Inc. Painfield, Illinois USA), converted in W m-2 and multiplied by the exposure time in a day. Two UV doses were tested: 8.5 kJ m^−2^ d^−1^ (30 min UV-B exposure, low dose) and 68 kJ m^−2^ d^−1^ (4 h UV-B exposure, high dose) while the control plants were exposed to cool white light at 100 μmol m^−2^ s^−1^ provided by fluorescence tubes (L 36 W/76 florescence tubes (OSRAM, Munich, Germany). UV-B doses were given once continuously in a single day (day zero). The Falcon tubes plants of low and high UV-B were placed at the same time inside the UV-B cabinet and arranged randomly within the shelf at 40 cm from the UV-B light source. Then, low and high dose irradiated plants were moved from the cabinet after 30 min and 4 h from the beginning of UV-B exposure, respectively, to the shelf where the control plants grew under a 16 h photoperiod, (100 µmol^−1^ m^−2^ s^−1^ white light), at a constant temperature of 22 °C, 65% RU, for 72 h. The UV-B treatments was repeated five times.

### 4.3. Chlorophyll a Fluorescence Transient Analysis and Parameters

The chlorophyll a fluorescence transient was measured using portable fluorimeter (Handy PEA - Hansatech Instruments Ltd., UK). Fully developed leaves from *O*. *basilicum*, which entirely filled the area of the sensor, were selected for the measurements, after adaptation in the dark for 30 min using leaf clips. After adaptation, the fluorescence parameters were measured using a saturating light pulse of 3000 µmol photon m-2 s-1, which closed all RCs. Chlorophyll a fluorescence transient obtained from the dark-adapted sample was analyzed with the OJIP-test [23,25,26]. The description and calculation formula of the parameters considered in this study are reported in Table 1.

### 4.4. Photosynthetic Pigments Quantification by HPLC

The extraction of the photosynthetic pigments was performed on disks of leaves frozen in liquid nitrogen and stored at -80 °C. Disks were extracted with 100% HPLC-grade methanol (1:10, w: v) overnight at 4 °C. HPLC grade solvents were used for the analyses of the extracts and the HPLC equipment (Jasco, Tokyo, Japan) included a PU-2089 four-solvent low-pressure gradient pump and a MD-4010 diode array detector. The HPLC separation was performed using a Macherey–Nagel C18 250/4.6 Nucleodur^®^ 100-5 Isis column equipped with a guard column, at a flow rate of 1 mL min-1. Methanol (solvent A), acetonitrile (solvent B), and ethyl acetate (solvent C) were used for elution, with the following gradient: 0–15 min, A 25%, B 75%; 15–17.5 min, A 70%, C 30%; 38 min, A 70%, C 30%; 41 min, A 25%, B 75%; 48 min, A 25%, B 75%. Detection was made in the wavelength range 270–700 nm. Standards of violaxanthin, neoxanthin, zeaxanthin, chlorophyll a and chlorophyll b (DHI, Hørsholm, Denmark) in a range of 0.78 to 5 mg L^−1^ and lutein and β-carotene (Sigma-Aldrich, Milano, Italy) in a range of 30 to 500 mg L^−1^ were used. Calibration curves, correlating the peak area to pigment concentration (R^2^ > 0.97), were performed.

### 4.5. Determination of Phenolic Acids by HPLC Analysis

The concentration of phenolic acids in each sample was determined using 300 mg frozen tissue, which was ground in a mortar with 5 mL extraction solvent (EtOH:H2O:HCl 80:19:1 v/v) and transferred into plastic tubes. Samples were sonicated for 1 h on 4 periods of time with ice and stored overnight at −20 °C, then they were centrifuged for 5 min at 3000 rpm. The supernatant was collected in plastic tubes and the pellet was extracted again with 5 mL fresh solvent. Samples were sonicated as in the previous extraction step and stored overnight at -20 °C. After centrifugation, the two supernatants were combined for analysis. All extracts were filtered with Chromafil^®^ 0.45 μm cellulose mixed esters membrane; 25 mm diameter syringe filters (Macherey-Nagel, Düren, Germany) prior to HPLC separations. HPLC grade solvents were used for the analyses, which were performed with the same apparatus used for pigments determination, at a flow rate of 1 mL min^−1^. Acetonitrile (solvent A) and aqueous 0.1% phosphoric acid (solvent B) were used for elution, with the following gradient: 0–0.4 min, A 5%; 0.4–10 min, A 5%–50%; 10-15 min, A 50%–95%; 15–17 min, A 95%; 17–19 min, A 95%–5%; followed by 5 min equilibration (A 5%). Detection was made in the wavelength range 220–450 nm. The injection volume was 20 μL and the analyses were performed at room temperature (23–24 °C). Phenolic acids were identified by comparing the retention times and absorption maxima of peaks obtained with analytically pure standards. For this purpose, ferulic, caffeic, chlorogenic, caftaric, p-coumaric and t-cinnamic acids (Sigma-Aldrich, Milano, Italy), and rosmarinic acid (Phytolab GmbH, Westenbergsgreuth, Germany) were used. The detection limit of the analytical protocol was on the order of 0.05 mg g^−1^ DW.

### 4.6. Hormone Analysis

ABA was determined by an indirect ELISA based on the use of DBPA1 monoclonal antibody raised against S (+)-ABA. Briefly, leaf samples (100 mg FW) were collected, weighted, frozen in liquid nitrogen, and then stored at −80 °C until analysis. ABA was measured after extraction in distilled water (water: tissue ratio = 10: 1 v: w) overnight at 4 °C according to the method previously described [65]. Ethylene production was measured by enclosing two Falcon tubes in each air-tight containers (1000 mL). Two mL gas samples were taken from the headspace of the containers after 1 h incubation at room temperature. The ethylene concentration in the sample was measured by a gas chromatograph (HP5890, Hewlett-Packard, Menlo Park, California) using a flame ionization detector (FID), a stainless steel column (150 × 0.4 cm ø packed with Hysep T), a column, and detector temperatures of 70 °C and 350 °C, respectively, as well as nitrogen carrier gas at a flow rate of 30 mL min^−1^.

### 4.7. Evans Blue Staining

To obtain a visual detection of cell death, the leaves were stained using Evans blue dye according to the methods previously described [66]. Leaves were taken at the end of the experiment (after 72 h) and soaked in 10 mL of 0.25% Evans blue. They were then washed briefly in 10 mL water. Leaves were de-stained in boiling 96% ethanol for 10 min. Then the leaves were transferred to a 60% glycerol solution.

### 4.8. Experimental Design and Statistical Analysis

To measure the biological variation in our data, we used 30 falcon-tubes (each falcon-tube contained two plants) for each treatment. The whole experiment counting the treatments and control was independently repeated five times. By repeating the whole UV-B treatment experiment independently several times, biological replicates were obtained. To obtain independent data from the experimental design, we performed and applied a formal procedure of statistical inference (e.g., statistical significance testing), and a biological replicate from each independent experiment was used for downstream analysis as described below. A fluorescence measurement using chlorophyll was carried out immediately after UV-B exposures (0 h), at 24, 48, and 72 h of recovery time (number of biological replicates, n=5). Pigments and phenolic acids were analyzed during UV-B exposures (0, 30 min and 4 h) and during recovery period of 72 h (24, 48, and 72 h), (n = 5, each replication was created by pooling together three leaves from different falcon-tube plants). Hormones analysis were performed during UV-B exposures (0, 15, 30 min and 4 h) and during a recovery period of 72 h (6, 24, 48, and 72 h), (n=5, each replication was created by pooling together three leaves from different Falcon-tube plants or by using plants from two Falcon tubes inside a container for ABA and ethylene, respectively). Data were analyzed within each sampling time using one-way ANOVA. Significant differences among means were estimated at the 5% (P < 0.05) level, using Tukey’s test (GraphPad Prism6, GraphPad Software, Inc., CA USA).

## Figures and Tables

**Figure 1 plants-08-00396-f001:**
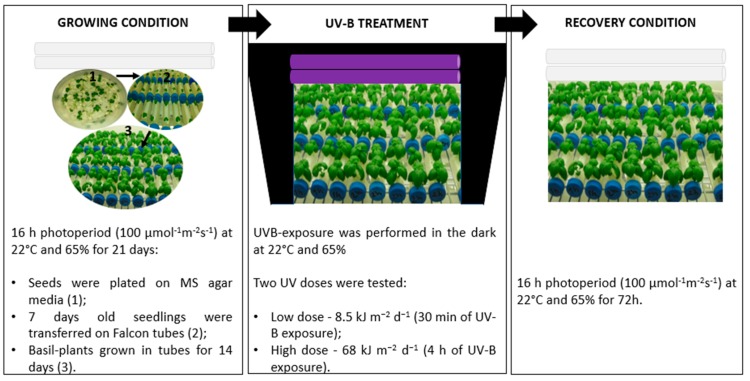
Schematic representation of overall experimental design from plant growing, UV-B irradiation to recovery conditions of sweet basil.

**Figure 2 plants-08-00396-f002:**
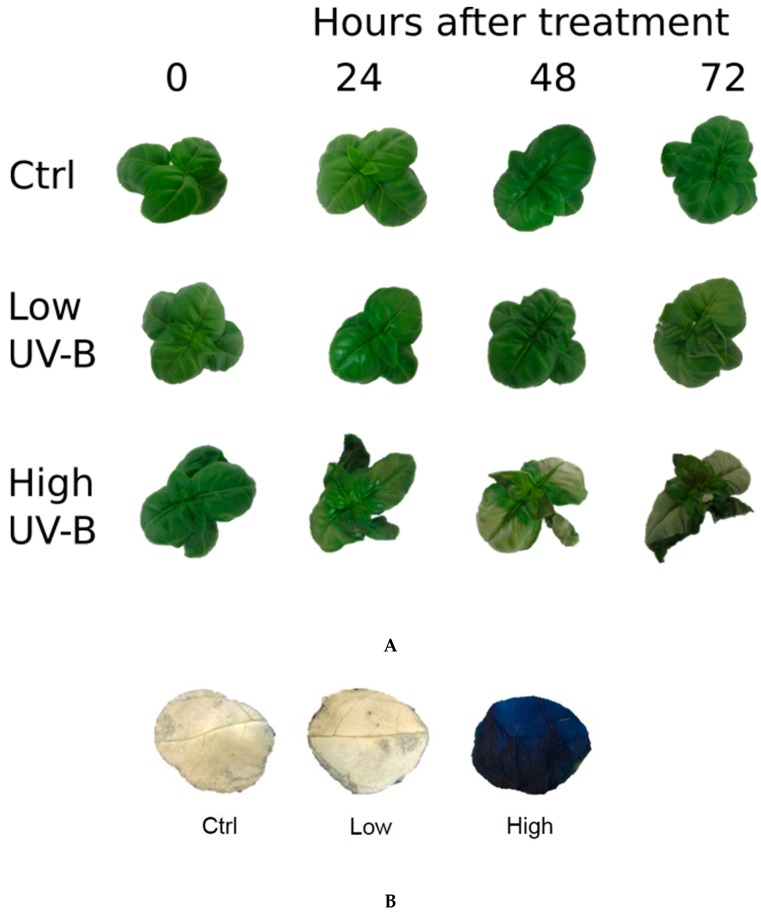
(**A**) Appearance of sweet basil plants over 72 h, after low and high UV-B exposures and control (Ctrl) condition (not UV-B exposed); (**B**) Leaf disks of sweet basil cv. Genovese irradiated with low UV-B light (30 min: 8.5 kJ m^−2^ day^−1^), high UV-B light (4hours: 68 kJ m^−2^ day^−1^) or not UV-B exposed (control). Cell death was monitored by staining the leaves with Evans blue after 72 h from the beginning of exposure. The pattern presented is representative of at least 10 replicates.

**Figure 3 plants-08-00396-f003:**
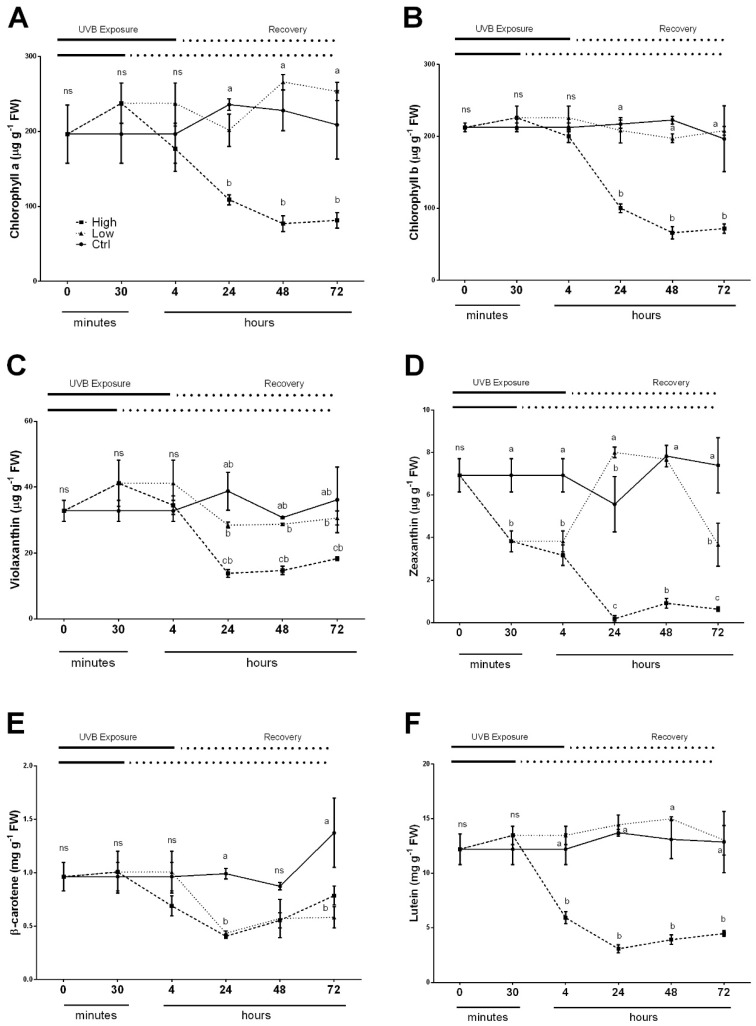
Time course of Chlorophyll a (**A**), Chlorophyll b (**B**), Violaxanthin (**C**), Zeaxanthin (**D**), β-Carotene (**E**) and Lutein (**F**) in sweet basil cv. Genovese irradiated with low UV-B light (30 min: 8.5 m^−2^ day^−1^), high UV-B light (4hours: 68 kJ m^−2^ day^−1^) or not exposed (control). Data are shown as mean with at least five independent biological replicates. The measurements are carried out at 0, 30 min, 4, 6, 24, 48 and 72 h from the beginning of exposure. Data were subjected to analysis of variance and differences between UV-B doses were analyzed by a Tukey’s multiple comparisons test. Different letters within the same time-point denote significant differences at P < 0.05. The thick line above the graph indicates the time of UV-B exposures (low: 30 min, high: 4 h, from bottom to top, respectively) and the dot line indicates the recovery period under normal light (low: 30 min, high: 4 h, from bottom to top, respectively).

**Figure 4 plants-08-00396-f004:**
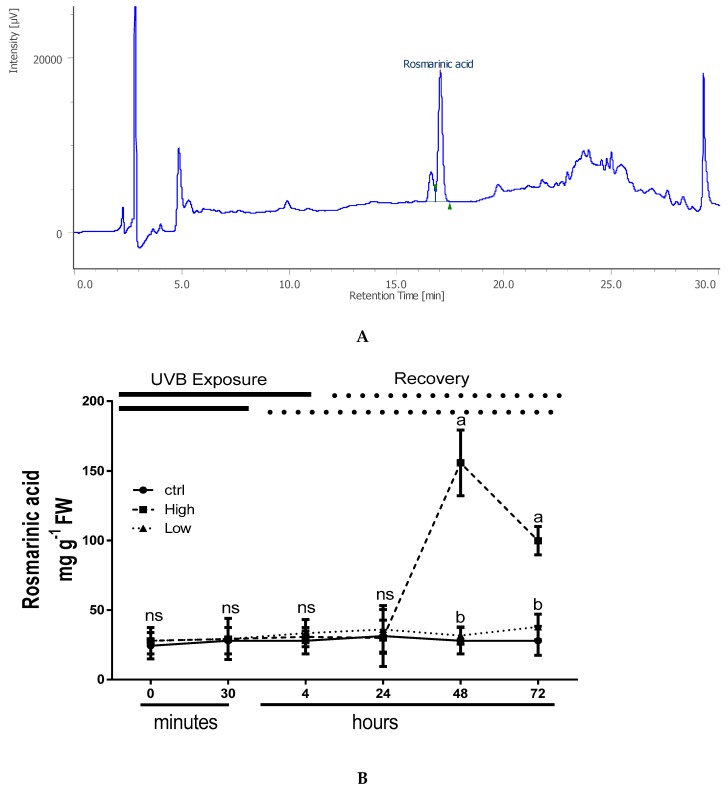
(**A**) Typical HPLC chromatogram of HCl-methanolic extract of sweet basil leaves (*O. basilicum* L.), showing a main peak that was identified as RA by means of LC-MS analysis; (**B**) Time course of changes in rosmarinic acid, in sweet basil cv. Genovese irradiated with low UV-B light (30 min: 8.5 kJ m^−2^ day^−1^), high UV-B light (4hours: 68 kJ m^−2^ day^−1^) or not-exposed plants (control). Data are shown as mean with at least 5 independent biological replicates. The measurements are carried out at 0, 30 min, 4 h, 24 h, 48 h, and 72 h from the beginning of exposure. Data were subjected to analysis of variance and differences between UV-B doses were analyzed by a Tukey’s multiple comparisons test. Different letters within the same time-point denote significant differences at P < 0.05. The thick line above the graph indicates the time of UV-B exposures (low: 30 min, high: 4 h, from bottom to top, respectively) and the dot line indicates the recovery period under normal light (low: 30 min, high: 4 h, from bottom to top, respectively).

**Figure 5 plants-08-00396-f005:**
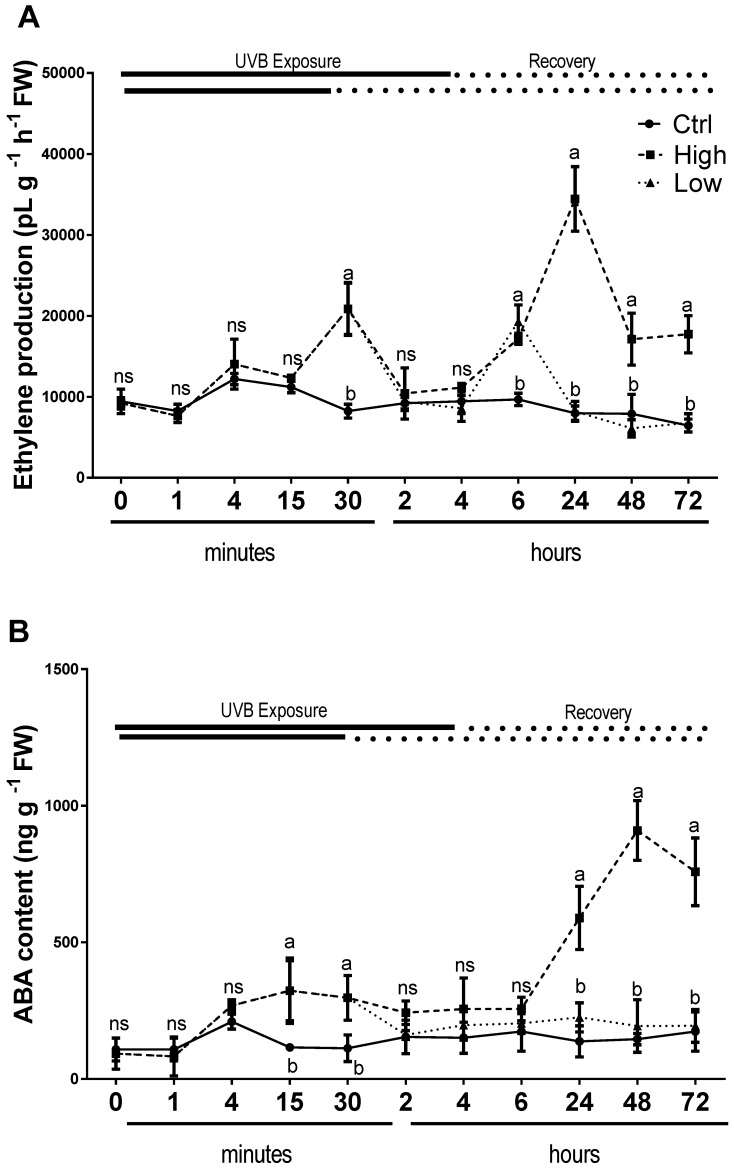
Time course of ethylene (**A**) emission and ABA content (**B**), in sweet basil cv. Genovese irradiated with low UV-B light (30 min: 8.5 kJ m^−2^ day^−1^), high UV-B light (4hours: 68 kJ m^−2^ day^−1^) or not exposed (control). Data are shown as mean with at least 5 independent biological replicates. The measurements are carried out at 0 1, 4, 15, and 30 min, as well as 2, 4, 6, 24, 48, and 72 h from the beginning of exposure. Data were subjected to analysis of variance and differences between UV-B doses were analyzed by a Tukey’s multiple comparisons test. Different letters within the same time-point denote significant differences at P < 0.05. The thick line above the graph indicates the time of UV-B exposures (low: 30 min, high: 4 h, from bottom to top, respectively) and the dot line indicates the recovery period under normal light (low: 30 min, high: 4 h, from bottom to top, respectively).

**Table 1 plants-08-00396-t001:** List of the OJIP test parameters with explanations and formulae used for calculation according to Kalaji et al. [25,26].

Parameter	Calculation	Description
***Extracted and technical fluorescence parameters***		
**Fo**	Fluorescence intensity at 50 µs	Fluorescence intensity when all reaction centers (RCs) are open
**Fj**	Fluorescence intensity at 2 ms at J-step	
**Fk**	Fluorescence intensity at 300 µs at K-step	
**Fm**	Maximal fluorescence intensity	Fluorescence intensity when all RCs are closed
**Vj**	Vj = (Fj − Fo)/(Fm − Fo)	Relative variable fluorescence at 2 ms. For unconnected PSII units, this equals the fraction of closed RCs expressed as a proportion of the total number of RCs
**Vk**	Vk = (Fk − Fo)/(Fm − Fo)	Relative variable fluorescence at 300 μs
**Fv/Fo**	(Fm − Fo)/Fo	Proportional to the activity of the water-splitting complex on the donor side of the PSII
**Fk/Fj**		To probe the extent of inactivation of the PSII donor side
**Mo**	Mo = 4(F300 − Fo)/(Fm − Fo)	Slope of the normalized curve at the origin of the fluorescence rise. Net rate of closed reaction centers accumulation
**Sm**	Sm = Area/(Fm−Fo)	Standardized area above the fluorescence curve between Fo and Fm is proportional to the pool size of the electron acceptors Q_A_ on the reducing side of Photosystem II
**Tfm**		Time needed to reach Fm
***Efficiencies and quantum yields***		
**ϕP_0_**	ϕP_0_ = 1 − (Fo/Fm) = Fv/Fm	Maximum quantum yield of primary PSII photochemistry. Probability that an absorbed photon will be trapped by the PSII RC with the resulting reduction of QA
**ϕE_0_**	ϕE_0_ = [1 − (Fo/Fm)](1 − Vj)	Quantum yield for electron transport
**ΨE_0_**	ΨE_0_ = 1-Vj	Efficiency of excitation energy to electron transport flux conversion. Probability that an exciton trapped by the PSII RC enters the electron transport chain
**δR_0_**	δR_0_ = (1 − Vi) (1 − Vj)	Efficiency with which an electron from the intersystem electron carriers moves to reduce end electron acceptor side (RE)
**δD_0_**	δD_0_ = 1 − ϕP_0_	It expresses the probability that the energy of an adsorbed photon is dissipated as heat
**ϕR_0_**	ϕR_0_ = δR_0_*ϕP_0_*ΨE_0_	Quantum yield for the reduction of end acceptors of PSI per photon absorbed

**Table 2 plants-08-00396-t002:** Effect of UV-B exposures (low: 8.5 kJ m^−2^ day^−1^; high: 68 kJ m^−2^ day^−1^; ctrl: not UV-B exposure) at different recovery time (0, 24, 48, and 72 h) on selected chlorophyll a fluorescence parameters of *O. basilicum* leaves. Means within each column (time after UV-B exposure) and within each chlorophyll a fluorescence parameter assessed followed by different letters are significantly different (P < 0.05; Tukey’s multiple range test, n ≥ 5). Arrows indicate the direction of the effects (P < 0.05) for each parameter under UV-B exposures (↑ red highlighted: the parameter has a higher value in UV-B treated leaves compared to ctrl ones; ↓ blue highlighted: the parameter has a lower value).

Parameter	UV-B dose	Time after UV-B exposure							
		0		24		48		72	
**Fo**	Ctrl	567.8 ns		569.5 b		614.7 b		626.0 b	
	Low	565.7 ns		609.5 b		668.0 b		749.0 bc	
	High	678.3 ns		816.5 a	↑	953.3 a	↑	907.0 ac	↑
Fm	Ctrl	3167 a		3011 a		3238 a		3198 a	
	Low	2641 a		2814 a		2341 b	↓	1849 b	↓
	High	1929 b	↓	1778 b	↓	1194 c	↓	862 c	↓
Vj	Ctrl	0.45 ns		0.49 ns		0.49 b		0.52 b	
	Low	0.43 ns		0.40 ns		0.47 b		0.48 b	
	High	0.45 ns		0.50 ns		0.67 a	↑	1.11 a	↑
Vk	Ctrl	0.30 ns		0.31 ns		0.31 ns		0.33 b	
	Low	0.28 ns		0.28 ns		0.29 ns		0.30 b	
	High	0.30 ns		0.29 ns		0.40 ns		0.67 a	↑
Fv/Fo	Ctrl	5,58 a		5,30 a		5,27 a		5.12 a	
	Low	4,68 a		4,61 a		3.54 b	↓	2.50 b	↓
	High	2,9 b	↓	2,41 b	↓	1.30 c	↓	1.00 c	↓
Fk/Fj	Ctrl	0.77 b		0.76 b		0.77 b		0.76 c	
	Low	0.78 b		0.82 b		0.80 b		0.84 b	↑
	High	0.86 a	↑	0.89 a	↑	0.95 c	↑	0.99 a	↑
Mo	Ctrl	1.19 ns		1.27 ns		1.27 b		1.35 b	
	Low	1.10 ns		1.10 ns		1.10 b		1.20 b	
	High	1.20 ns		1.17 ns		1.70 a	↑	2.67 a	↑
Sm	Ctrl	17.83 b		19.76 ns		20.25 ns		20.25 a	
	Low	31.34 b		33.50 ns		32.14 ns		28.76 a	
	High	71.74 a	↑	41.33 ns		13.33 ns		0.33 b	↓
Tfm	Ctrl	195.0 b		205.0 b		205.0 ns		210.0 a	
	Low	260.0 b		280.0 b		390.0 ns		285.0 a	
	High	750.0 a	↑	510.0 a	↑	224.8 ns		1.5 b	↓
ϕP_0_	Ctrl	0.82 ns		0.81 a		0.81 a		0.80 a	
	Low	0.78 ns		0.78 a		0.69 a		0.60 b	↓
	High	0.62 ns		0.46 b	↓	0.18 b	↓	0.01 c	↓
ϕE_0_	Ctrl	0.45 ns		0.41 a		0.41 a		0.41 a	
	Low	0.45 ns		0.47 a		0.37 a		0.32 a	
	High	0.35 ns		0.26 b	↓	0.08 b	↓	0.001 b	↓
ΨE_0_	Ctrl	0.55 ns		0.51 ns		0.51 a		0.51 a	
	Low	0.57 ns		0.60 ns		0.52 a		0.52 a	
	High	0.55 ns		0.49 ns		0.33 b	↓	0.03 b	↓
δR_0_	Ctrl	0.48 b		0.46 ns		0.44 ns		0.49 a	
	Low	0.59 b		0.54 ns		0.52 ns		0.49 a	
	High	0.75 a	↑	0.54 ns		0.33 ns		0.17 b	↓
ϕR_0_	Ctrl	0.14 ns		0.15 ns		0.15 a		0.15 a	
	Low	0.20 ns		0.20 ns		0.20 a		0.09 a	
	High	0.17 ns		0.10 ns		0.02 b	↓	0.000 b	↓
δD_0_	Ctrl	0.18 ns		0.19 b		0.19 b		0.19 c	
	Low	0.21 ns		0.22 b		0.31 b		0.40 b	↑
	High	0.37 ns		0.53 a	↑	0.82 a	↑	0.99 a	↑

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
