# Peer review of "UV-B Physiological Changes Under Conditions of Distress and Eustress in Sweet Basil"

_plants, 2019, doi:10.3390/plants8100396_

Round 1
Reviewer 1 Report
The current version of the paper entitled: " UV-B physiological changes under conditions of distress and eustress in sweet basil " is well presented and structured and all the experiments have been carried out properly and the data analyzed and interpreted as expected.
1. I suggest a thorough revision of the formatting of the references section according to the journal's guidelines.
2. The authors should correct “Ocium basilicum L.” in “Ocium basilicum L.” according to the Systematic Botany classification. Please, check the entire manuscript.
Author Response
Please see the attachment
our responses to reviewers and editor
Best Regards
Alice Trivellini

Reviewer 2 Report
Review of the Manuscript ID: Plants-603894, titled: "UV-B physiological changes under conditions of distress and eustress in sweet basil "
This manuscript is interesting, well-written and provides information on the specific response mechanisms regulating photoprotection vs irreversible photoinhibition in basil depending on the UV-B doses. I would like to recommend this paper for publication after minor revision.
The satisfactory clarification of some points is needed:
I think that Table 1 should be given in Materials and Methods (Section 4.3.), as this table contains description and calculation formulae of the fluorescence parameters considered in this study. The authors should also give references for the description of fluorescence parameters. The authors argue that there is a direct correlation between the parameter Fv/Fo and “the efficiency of the water-splitting complex on the donor side of PSII”, but nowhere in the manuscript they do not give references about it. Whether the authors have previous data obtained by measuring the activity of oxygen evolving complex?Author Response
Please see the attachments
the revised manuscript our responses to reviewers and editor
Best Regards
Alice Trivellini
